# Characterization of Mechanical Heterogeneity in Dissimilar Metal Welded Joints

**DOI:** 10.3390/ma14154145

**Published:** 2021-07-26

**Authors:** He Xue, Zheng Wang, Shuai Wang, Jinxuan He, Hongliang Yang

**Affiliations:** 1School of Mechanical Engineering, Xi’an University of Science and Technology, Xi’an 710054, China; 19205201083@stu.xust.edu.cn (Z.W.); 17101016005@stu.xust.edu.cn (S.W.); hjx84118631@163.com (J.H.); 2Center of Engineering Training, Xi’an University of Science and Technology, Xi’an 710054, China; hl_yang@163.com

**Keywords:** dissimilar metal welded joints, mechanical heterogeneity, UMAT, numerical simulations, integrity assessment

## Abstract

Dissimilar metal welded joints (DMWJs) possess significant localized mechanical heterogeneity. Using finite element software ABAQUS with the User-defined Material (UMAT) subroutine, this study proposed a constitutive equation that may be used to express the heterogeneous mechanical properties of the heat-affected and fusion zones at the interfaces in DMWJs. By eliminating sudden stress changes at the material interfaces, the proposed approach provides a more realistic and accurate characterization of the mechanical heterogeneity in the local regions of DMWJs than existing methods. As such, the proposed approach enables the structural integrity of DMWJs to be analyzed in greater detail.

## 1. Introduction

The leading pressure-bearing equipment in the primary water systems of pressurized water reactors (PWRs) is connected to the main pipeline using a safe-end dissimilar metal welded joint (DMWJ). Because of their high corrosion resistance and suitable mechanical properties, nickel-based alloys are widely used as welding metals for connecting stainless steel piping and pressure vessel nozzles [1,2]. In general, DMWJs consist of at least three base materials and weld materials and are subjected to a variety of complex loads while in service; therefore, structural failures often occur at DMWJs [1,2,3]. Given their status as a critical region for structural integrity analysis, the mechanical heterogeneity in the locality of DMWJs needs to be expressed appropriately. However, current approaches for assessing the structural integrity of DMWJs have certain limitations and rely heavily on the use of general material strength analysis and the operational experience of the inspector [4]. As such, a more accurate expression of the mechanical heterogeneity of DMWJs is required to assist in the analysis of their local mechanical properties [5].

The highly heterogeneous distribution of fracture properties, mechanical properties, and microstructure along the DMWJ has restricted the development of mechanical property evaluation systems [6,7]. Existing structural integrity assessments of welded joints, such as those proposed by Fan et al. [8,9] and Xue et al. [10], tend to ignore the influence of the heat-affected zone (HAZ) and the fusion zone (FZ). The simple bi- or tri-material models typically used do not evaluate the mechanical properties of strength-mismatched welded joints directly, potentially affecting the accuracy of calculations. Engineering applications that consider the mechanical heterogeneity of DMWJs usually use sandwich composite structures [11,12,13]. For simplicity, the welded joints are partitioned, with each partition corresponding to the mechanical parameters of a specific material. Despite their outstanding structural properties, such sandwich composite structures are mismatched in terms of their interregional material and geometric properties [14,15]. Significant interfacial stress variations, induced by mechanical loadings, occur at each regional interface [16,17]. Consequently, sandwich composite structures retain several limitations in terms of expressing the mechanical properties of the materials in the locality of the DMWJ [18,19]. To address the abrupt changes in the interfacial stresses encountered in simplified sandwich composite structures, a temperature field can be introduced to analyze the mechanical behavior in the locality of the DMWJ. To approximate the mechanical heterogeneity in terms of the temperature field, different temperatures can be defined at different locations [20,21]. By reproducing this method, we found that it yielded an approximation of the mechanical heterogeneity in the local regions of the DMWJ. To represent the mechanical heterogeneity at arbitrary positions in a simple and robust manner, the mechanical parameters were defined for the different element integration points [22].

This objectives of this study were: (i) to develop a more appropriate method for characterizing the mechanical heterogeneity of the DMWJ based on the User- defined Materials (UMAT) subroutine and (ii) to elucidate the variation of mechanical properties in the local regions of the DMWJ. The heat-affected and fusion zones of mechanical heterogeneity were considered in the analysis of the changes in the mechanical behavior of local regions of the DMWJ. Numerical simulations of the stress–strain curves at different positions in the vicinity of the DMWJ interface indicated that the interface location was likely to be a weak point in DMWJ failure. The proposed method can be applied to analyze the integrity of important engineering structures and to improve the stability of vulnerable structures.

## 2. Materials and Methods

### 2.1. UMAT Theoretical Basis

The UMAT subroutine in the finite element analysis (FEA) software ABAQUS (Abaqus 6.14, Dassault Systemes, Paris, France) can be used to define the mechanical behavior of materials and perform calculations at different element integration points [23]. In this study, the mechanical behavior of the element integration points was matched, point-by-point, to the coordinate positions in order to determine the mechanical properties at different positions in the DMWJ more realistically. Compared with previous methods, this approach was simpler and offered better accuracy and reproducibility. The UMAT subroutine calculated the trial and equivalent stresses, the equivalent plastic strain, and updated the state variables.

When an external force is applied, the subroutine calculates the elastic trial stress $\sigma_{n+1}^{trial}$ at the moment (*n* + 1) based on the total strain increment ∆*ε* applied at the moment $t_{n}$; thus, the total strain increment ∆*ε* is decomposed as
(1)$$∆\varepsilon =∆\varepsilon_{e}+∆\varepsilon_{p},$$
where $∆\varepsilon_{e}$ and $∆\varepsilon_{p}$ are the elastic and plastic components of the total strain, respectively.

Based on the assumption that the initial state is purely elastic, the trial stress at the beginning of the subroutine can be expressed as
(2)$$\sigma_{n+1}^{trial}=\sigma_{n}+D_{e}:∆\varepsilon_{e},$$
where *trial* represents the trial state and $D_{e}$ is the Jacobian matrix in the elastic state:(3)$$D_{e}=\frac{E}{2\left(1+v\right)}\left(\delta_{ik}\delta_{jl}+\delta_{il}\delta_{jk}\right)+\frac{vE}{\left(1+v\right)\left(1-2v\right)}\left(\delta_{ij}\delta_{kl}\right),$$
where $\delta_{ij}$ denotes the abbreviation for the unit matrix (when i=j, $\delta_{ij}=1$; when i≠j, $\delta_{ij}=0$), *E* is Young’s modulus, and *v* is Poisson’s ratio.

Next, the equivalent stress $\bar{\sigma }$ can be expressed as [24]
(4)$$\bar{\sigma }=\sqrt{\frac{3}{2}}∥S∥,$$
where *S* represents deviatoric stress.

According to the $J_{2}$ flow rule [22],
(5)$$∆\varepsilon_{p}=∆\lambda \frac{3S}{2\bar{\sigma }},$$
which means that the plastic strain $\varepsilon_{n+1}^{p}$ at the moment $t_{n+1}$ is
(6)$$\varepsilon_{n+1}^{p}=\varepsilon_{n}^{p}+1.5∆\lambda \frac{s_{n+1}^{trial}}{{\bar{\sigma }}_{n+1}^{trial}}.$$

According to Ref. [24], the equivalent plastic strain increment at the *n*th step is
(7)$$∆\varepsilon_{n}^{p}=∆\lambda =\frac{f_{n+1}^{trial}}{H+3G}.$$

Updating the stress values in the UMAT subroutine necessitates using different algorithms for the elastic and plastic stages. When the stress lies within the elastic range, the relationship between the stress and strain increments can be expressed as
(8)$$d\sigma =D_{e}\left(d\varepsilon -d\varepsilon_{p}\right),$$
whereas when the stress is beyond the elastic range
(9)$$d\sigma =D_{ep}d\varepsilon ,$$
where $D_{ep}=D_{e}-D_{p}\;\mathrm{in}\;\mathrm{which}\;D_{p}=\frac{9G^{2}*S*S^{T}}{{\bar{\sigma }}^{2}\left(H^{\prime }+3G\right)}$ [24,25].

The integration algorithm used by the UMAT subroutine comprises two main parts: the elastic trial and the return mapping. When loaded by an external force, the elastic trial stress $\sigma_{n+1}^{trial}$ at the moment $t_{n+1}$ and the difference between it and the yield function $f_{n+1}^{trial}$ are calculated by the subroutine according to the total strain increment Δ*ε* applied at the moment $t_{n}$, with the outcome determining whether the stress enters the plastic phase. When $f_{n+1}^{trial}\leq 0$, the integration algorithm is in the elastic phase ⓐ. In addition, because the plastic strain increment Δ*λ* = 0, the total strain increment Δ*ε* represents the elastic strain increment and the elastic stiffness matrix $D_{e}$ is used to update the stress at time $t_{n}$ to that at time $t_{n+1}$. Conversely, when $f_{n+1}^{trial}>0$, the integration algorithm is in the plastic phase ⓑ; therefore, the plastic strain increment Δ*λ* > 0, the total strain increment Δ*ε* is decomposed into elastic strain increment $∆\varepsilon_{e}$ and plastic strain increment $∆\varepsilon_{p}$, and the stress entering the plastic stage uses the elastic–plastic stiffness matrix $D_{ep}$ to update the stress at time $t_{n}$ to that at time $t_{n+1}$. Figure 1 shows the corresponding algorithm flow diagram.

### 2.2. Material Model

The DMWJs were modelled using Alloy52M as the filler metal and low-alloy steel (LAS) SA508 and austenitic stainless steel 316L as the base metal [26]. The chemical compositions of various materials are listed in Table 1 [26]. The DMWJ sample is shown in Figure 2. The weld metal consisted of buttering Alloy52Mb (52Mb) and weld Alloy52Mw (52Mw). The inner wall of the tube possessed an overlayer of austenitic stainless steel 304. This study focused mainly on the base metal (SA508 and 316L) and weld (Alloy52M) zones, where the mechanical heterogeneity was clearest.

Table 2 lists the room-temperature mechanical properties of various DMWJ materials according to Refs. [20,27].

When $\sigma \leq \sigma_{y}$, the material deforms elastically, whereas when $\sigma >\sigma_{y}$, the material deforms plastically; thus
(10)$$\sigma =\{\begin{array}{cc} E\varepsilon_{e}, & \sigma \leq \sigma_{y}\\ \sigma_{y}+H^{\prime }\left(\varepsilon_{P}\right), & \sigma >\sigma_{y} \end{array},$$
where σ is the stress, $\varepsilon_{e}$ is the elastic strain, $\sigma_{y}$ is the yield strength of the material, and *H*′ is the material hardening coefficient in the plastic deformation phase.

The continuous transition model developed in this study considers the continuous variation of the mechanical properties of the materials along the DMWJ. Taking the axial direction as the *y*-direction, the yield strength $\sigma_{y}$ of the welded joint varies along the axial direction according to the function $\sigma_{y}\left(y\right)$; upon entering the plastic deformation stage, the hardening coefficient $H^{\prime }$ changes according to the function $H^{\prime }\left(y\right)$. The yield strength $\sigma_{y}$ at different positions and the plastic-phase hardening coefficient *H*′ are expressed as
(11)$$\{\begin{array}{c} \sigma_{y}=\sigma_{y}\left(y\right)\\ H^{\prime }=H^{\prime }\left(y\right) \end{array}.$$

Substituting Equation (11) into Equation (10) yields the principal equation for the continuous variation of the mechanical properties along the DMWJ:(12)$$\sigma =\{\begin{array}{cc} E\varepsilon_{e}, & \sigma \leq \sigma_{y}\\ \sigma_{y}\left(y\right)+H^{\prime }\left(y\right)\left(\varepsilon_{P}\right), & \sigma >\sigma_{y} \end{array}.$$

In the UMAT subroutine, the mechanical heterogeneity of the DMWJ is characterized by defining *E*, *v*, $\sigma_{y}\left(y\right)$, and $H^{\prime }\left(y\right)$. at different positions in the DMWJ to establish a finite element model and then analyzing it.

Following the approach described in [28], the relationships between hardness and the yield and ultimate strengths in the base metal, HAZ, and weld metal are described by Equation (13). In addition, Equation (14) describes the plastic-phase hardening coefficient:(13)$$\{\begin{array}{cc} \sigma_{y}=3.28HV-221 & \left(\mathrm{BM},\;\mathrm{HAZ}\right)\\ \sigma_{u}=3.29HV-47 & \left(\mathrm{BM},\;\mathrm{HAZ}\right)\\ \sigma_{y}=3.15HV-168 & \left(\mathrm{WM}\right)\\ \sigma_{u}=2.84HV+28 & \left(\mathrm{WM}\right) \end{array},$$
(14)$$H=\frac{\sigma_{u}-\sigma_{y}}{∆\varepsilon_{p}}.$$

According to References. [3,11,27], the yield strengths of the DMWJ constituent materials (SA508, Alloy52M, and 316L) at room temperature were determined to be 426 MPa, 400 MPa, and 345 MPa, respectively. However, the local mechanical properties of the HAZ and the FZ during welding differed substantially from those of the base material (BM) and the weld metal (WM). Therefore, for the interface region, considering only the mechanical properties of the BM or WM may produce nonconservative (unsafe) or overly conservative results. By applying the empirical conversion equation to microhardness experimental data [28], the yield strength and hardening coefficient distributions at different positions along the DMWJ can be obtained, as shown in Figure 3.

Combined with the yield strength and hardening coefficient distributions along the DMWJ, this study used a polynomial function with a curve fitting correlation coefficient of $R^{2}\geq 0.95$ to fit the individual regions of the yield strength and hardening coefficient distributions obtained using the continuous transition material model. The continuous changes in the mechanical properties of the materials in the DMWJ interface region were considered in the constitutive equation and integrated into the UMAT subroutine:(15)$$\sigma_{y}=\{\begin{array}{cc} 426 & y\in \left[0,\;6.0\right]\\ 429.7-3.5y+0.5y^{2} & y\in \left(6.0,\;7.5\right]\\ -11,077+2913y-184y^{2} & y\in \left(7.5,\;8.5\right]\\ 400 & y\in \left(8.5,\;11.5\right]\\ -12,414+2139y-89.1y^{2} & y\in \left(11.5,\;12.5\right]\\ 400 & y\in \left(12.5,\;18.0\right]\\ -61,463+6724.5y-182.65y^{2} & y\in \left(18.0,\;19.0\right]\\ 365 & y\in \left(19.0,\;27.0\right] \end{array}.$$
(16)$$H=\{\begin{array}{cc} 1100 & y\in \left[0,\;6.0\right]\\ 884.1+60.8y-4y^{2} & y\in \left(6.0,\;7.5\right]\\ -49,186+12,719y-801.3y^{2} & y\in \left(7.5,\;8.5\right]\\ 1000 & y\in \left(8.5,\;11.5\right]\\ -63,069.14+10,697y-445.7y^{2} & y\in \left(11.5,\;12.5\right]\\ 1000 & y\in \left(12.5,\;18.0\right]\\ -249,904.163+27,253.1y-739.7y^{2} & y\in \left(18.0,\;19.0\right]\\ 74,094.72-10,621.2y+512.34y^{2}-8.22y^{3} & y\in \left(19.0,\;23.0\right]\\ 850 & y\in \left(23.0,\;27.0\right] \end{array}.$$

For analyses using the sandwich structure, the model was partitioned, with different regions assigned different yield strengths as per Equation (17):(17)$$\sigma_{y}=\{\begin{array}{cc} 426 & y\in \left[0,\;8.0\right]\\ 400 & y\in \left[8.0,\;18.5\right]\\ 365 & y\in \left[18.5,\;27.0\right] \end{array}.$$

### 2.3. Geometric Model

The DMWJ consisted of a base metal (A508 and 316L) away from the interface and a weld metal (buttering Alloy52Mb and weld Alloy52Mw). The transition from the base metal to the weld zone inevitably caused lattice distortion. Therefore, a narrow (0.5–1 mm) fusion zone (FZ) occurred near the interface—the composition, microstructure, and hardness of which all changed significantly. In addition, the structure and properties of the HAZ (1–5 mm) were unevenly distributed, owing to the heat-related effects of welding [3,11,17]. Figure 4 shows a simplified schematic diagram of the structure used for the FEA. It is important to note that material interfaces will not be perfectly straight in reality; however, for simplicity, straight interfaces were used in all simulations.

### 2.4. Mesh and Load Model

Although welding is a three-dimensional process, it is widely acknowledged that axisymmetric models are appropriate for simulating the welding of cylindrical structures to determine the local mechanical properties of DMWJs [29]. Therefore, axisymmetric finite element models, which are fast and easy to use, are widely used to simulate heterogeneity. As shown in Figure 5, the mesh was composed of 4-node bilinear axisymmetric elements (CAX4) with a total of 1080 elements (0.5 mm × 0.5 mm). Displacement and rotation constraints were applied to the left end of the model in the *y*-direction and around the *x*-direction, respectively. The mechanical properties of the model were compared and analyzed by applying $U_{y}$ = 0.5 mm, pressure = 0 MPa, and $U_{y}$ = 0.5 mm, pressure = 100 MPa, respectively.

## 3. Results and Discussion

### 3.1. Continuous Transition Model Simulation Results

The numerical simulation results for the model subjected to axial loading are shown in Figure 6. Specifically, Figure 6a,b show the stress results for the developed continuous transition and sandwich models, respectively. The results obtained for applying the continuous transition model revealed that the stress in the interface region gradually increased at first before gradually decreasing. In contrast, for the sandwich composite structure model, the stress at the interface changed suddenly. As shown in Figure 7, the models were subjected to axial loading and internal pressure simultaneously. As a consequence, the internal stresses of the model changed, although they remained most significant at the interface region. This highlights the importance of the interface region when assessing the structural integrity of DMWJs. The next section discusses the stress–strain relationship in the interface region for a representative example in which an axial loading of 0.5 mm was applied.

### 3.2. Local Stress–Strain Simulation Results at the SA508/52Mb and 52Mw/316L Interfaces

Figure 8a shows the stress–strain curves in the vicinity of the SA508/52Mb interface in response to an axial loading of 0.5 mm. The curves indicate the presence of high stress and low strain in the HAZ of the SA508 material. In contrast, the 52Mb region exhibited low stress and high strain. Figure 8b shows the stress–strain curves in the vicinity of the 52Mw/316L interface: compared with the 52Mw region, the 316L region, which was situated farther from the interface, exhibited lower stress and higher plastic strain. Nevertheless, both interface regions showed high stress and plastic strain, which promotes crack initiation and growth. Therefore, from the numerical simulation results, we inferred that the interface represents a high-probability zone for DMWJ failure, which is consistent with experimental results reported in the literature [27,30]. Figure 8 shows that the maximum stress at position e in the SA508/52Mb interface zone was 485 MPa, while the stress at point a in the base material was 435 MPa, representing a difference of 50 MPa. The maximum stress at position d in the 52Mb/316L interface region was 450 MPa, while the maximum stress at point a in the 52Mw region was 415 MPa, representing a stress change of 35 MPa. Owing to the larger magnitude of the stress change in the SA508/52Mb interface region, it was deemed more susceptible to destruction. As such, it should be focused on.

### 3.3. Effect of Mechanical Heterogeneity on the DMWJ Local Region Mechanical Properties

From Figure 9, it can be seen that the stress component in the *x*-direction was compressed in both the sandwich structure model and in the continuous transition model—a phenomenon most likely due to the axial loading-induced displacement. Moreover, this phenomenon was amplified by the effects of internal pressure. It was evident that the stress in the interface region changed significantly.

As shown in Figure 10, when axial loads alone were considered, the *y*-directional stress component was significantly larger than when both the axial and the internal pipe pressure loads were considered. This is due to the reduction of the *y*-directional stress component by the internal pressure. The *y*-directional component exhibited the same behavior under both loading conditions, with the stress in the interface region changing abruptly. This further supports the need to prioritize the mechanical properties of the interface region.

As shown in Figure 11, under an axial load, the sandwich structure model showed a significant increase in stress at the SA508/52M and 52M/316L interfaces. For an internal pressure loading, the opposite trend was observed and a sudden change in stress occurred at the interface region. The continuous transition model eliminated abrupt changes in stress, allowing for a continuous transition in the mechanical properties of the materials in the interface regions.

As shown in Figure 12, Figure 13, Figure 14 and Figure 15, by combining the analysis results for the stress components in the *x*-, *y*-, and *z*-directions, it was concluded that the sandwich structure model exhibited an abrupt change in stress and in the equivalent plastic strain (PEEQ) in the interface region. This can be attributed to the fact that the model assigns different mechanical properties for different partitions, while ignoring the existence of the HAZ and FZ. As shown in Figure 12, the results upon applying the developed continuous transition model revealed that, in the SA508/52M interface region, the stress gradually increased to 485 MPa before gradually decreasing to 405 MPa, whereas in the 52Mw/316L interface region, the stress gradually increased to 450 MPa before gradually decreasing to 376 MPa. In the sandwich structure model, the stress values did not transition continuously in the interface region but instead changed abruptly. This was evident in the sudden stress changes (from 430 MPa to 404 MPa in the SA508/52Mb interface region and from 404 MPa to 377 MPa in the 52Mw/316L interface region). For an internal pressure of 100 MPa, the analysis method was the same. In addition, the method proposed in this study can characterize the variation of mechanical properties between overlay layers composed of 52Mb and 52Mw. Thus, the method proposed herein is more appropriate for expressing the mechanical properties in the interface region and better for assessing the structural integrity of DMWJs.

The mechanical property variations shown in Figure 12, Figure 13, Figure 14 and Figure 15 are in general agreement with the room temperature stress–strain results obtained for different regions of welded components in previous studies [3,7,11]. Sudden changes in stress in the interface regions caused subsequent changes in the PEEQ. Therefore, DMWJs possess significant mechanical heterogeneity in multi-material interface regions, and the use of conventional sandwich structure models to study DMWJs may produce nonconservative (unsafe) or overly conservative results. Consequently, the mechanical property distributions for DMWJ interface regions must be treated properly when assessing the structural integrity of DMWJs.

## 4. Conclusions

This paper presented an approach for characterizing the mechanical heterogeneity in the local regions of DMWJs. Variations of mechanical properties in the local regions of the DMWJ composed of Alloy52M were analyzed, leading to the following conclusions: (1)The use of a user-defined material utility, UMAT, enables the continuous variation of mechanical properties in all regions of the DMWJ to be characterized to obtain more accurate and realistic results.(2)By considering the local mechanical heterogeneity, the proposed method avoids mismatching interregional material and geometric properties and eliminates abrupt changes in interfacial stresses. Accurate finite element simulations were performed using ABAQUS commercial software.(3)Owing to their serious mechanical heterogeneities, particularly in the interface regions, the local mechanical properties of DMWJs change considerably. Compared with the 52Mw/316L interface region, the SA508/52Mb interface region is more susceptible to mechanical heterogeneity and should be regarded as a high-risk region in terms of structural integrity.

Finally, the proposed method can also be used to analyze the micromechanical fields and growth paths of cracks at different positions in mechanically heterogeneous DMWJ interface zones. Such analyses will be the subject of future research.

## Figures and Tables

**Figure 1 materials-14-04145-f001:**
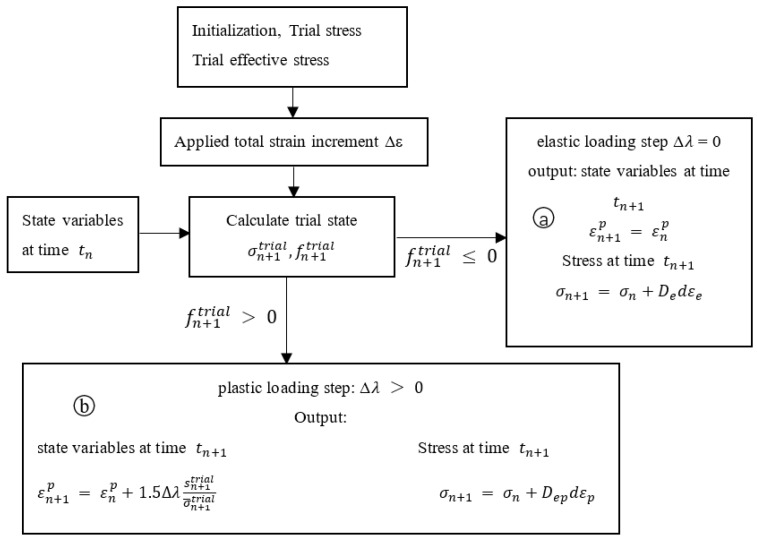
Schematic diagram of subroutine flow.

**Figure 2 materials-14-04145-f002:**
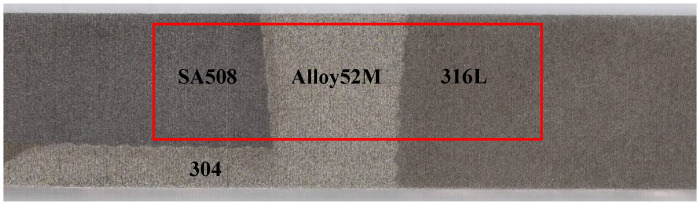
Physical diagram of dissimilar metal welded joints and sampling area.

**Figure 3 materials-14-04145-f003:**
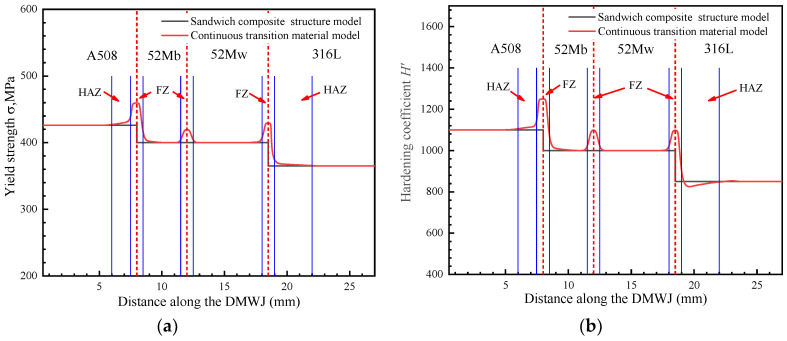
Distributions of (**a**) yield strength and (**b**) hardening coefficient along the DMWJ.

**Figure 4 materials-14-04145-f004:**
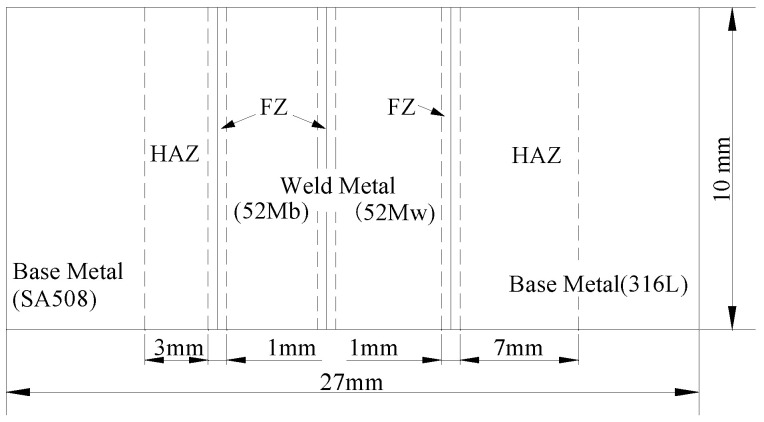
Schematic diagram of each region in the simplified welded structure.

**Figure 5 materials-14-04145-f005:**
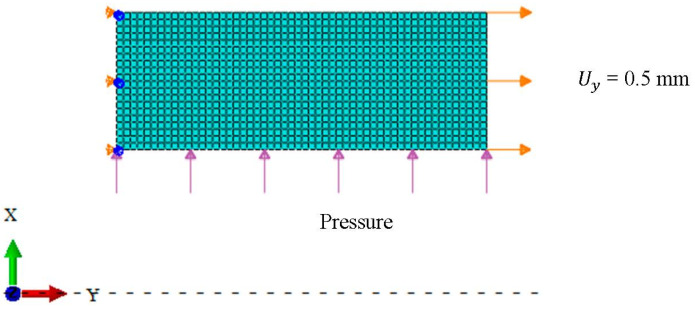
Finite element model of dissimilar metal welded joints.

**Figure 6 materials-14-04145-f006:**
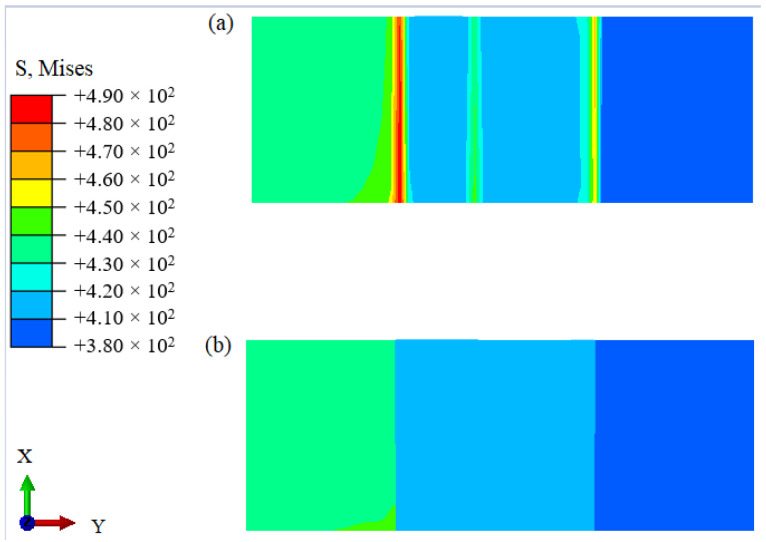
Numerical simulation results under an axial load: (**a**) continuous transition model, (**b**) sandwich model.

**Figure 7 materials-14-04145-f007:**
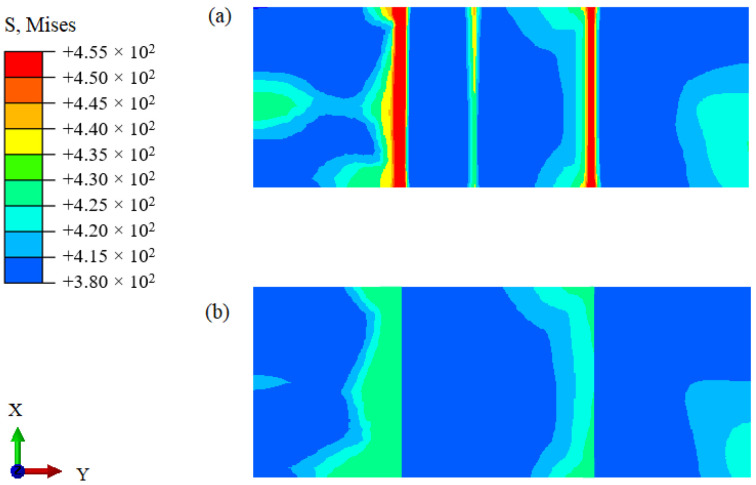
Numerical simulation results under the simultaneous action of axial load and internal pressure: (**a**) continuous transition model, (**b**) sandwich model.

**Figure 8 materials-14-04145-f008:**
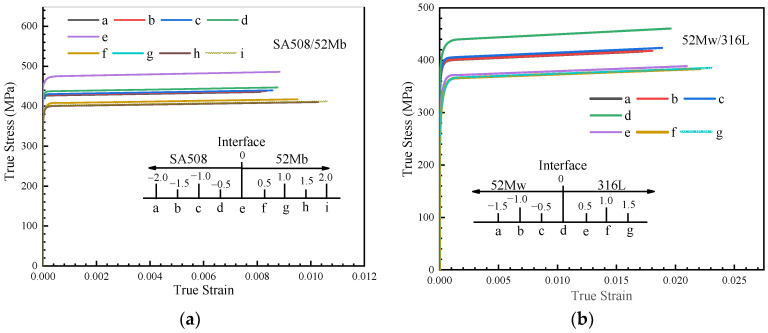
Stress–strain curves at the (**a**) SA508/52Mb and (**b**) 52Mw/316L interface zones.

**Figure 9 materials-14-04145-f009:**
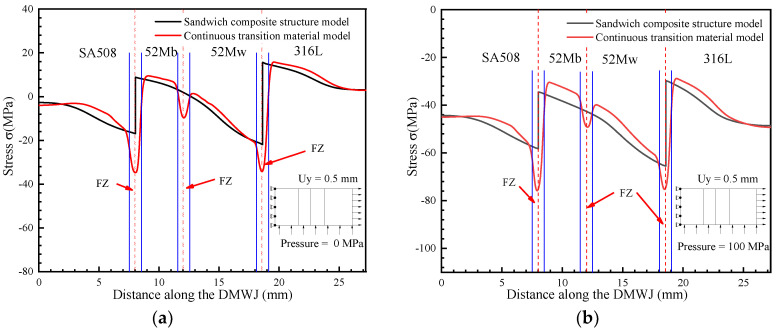
*X*-directional stress under (**a**) an axial load and (**b**) an axial load and internal pipe pressure.

**Figure 10 materials-14-04145-f010:**
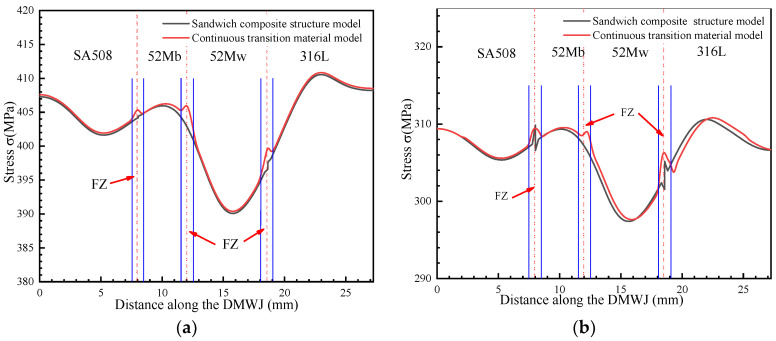
*Y*-directional stress under (**a**) an axial load and (**b**) an axial load and internal pipe pressure.

**Figure 11 materials-14-04145-f011:**
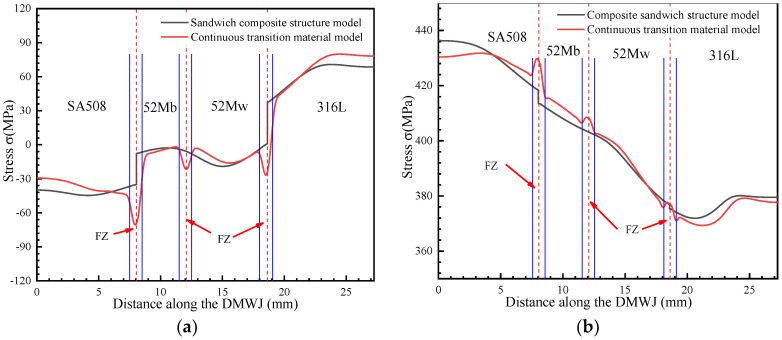
*Z*-direction stress under (**a**) an axial load and (**b**) an axial load and internal pipe pressure.

**Figure 12 materials-14-04145-f012:**
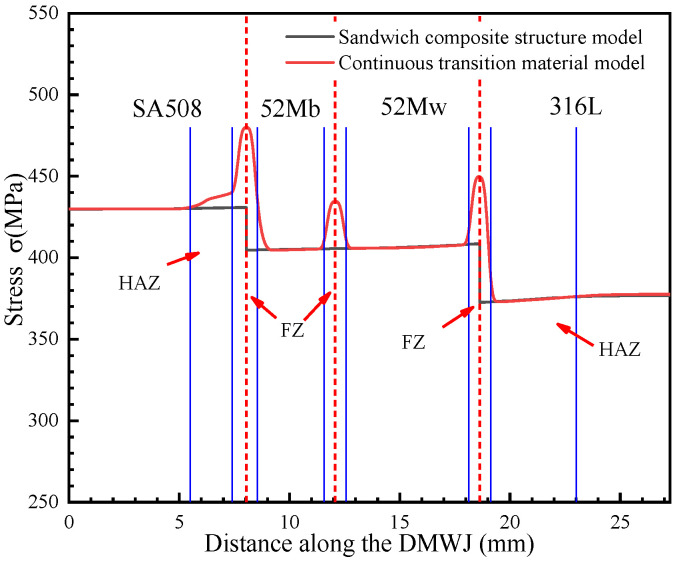
Stress along the DMWJ under an axial load.

**Figure 13 materials-14-04145-f013:**
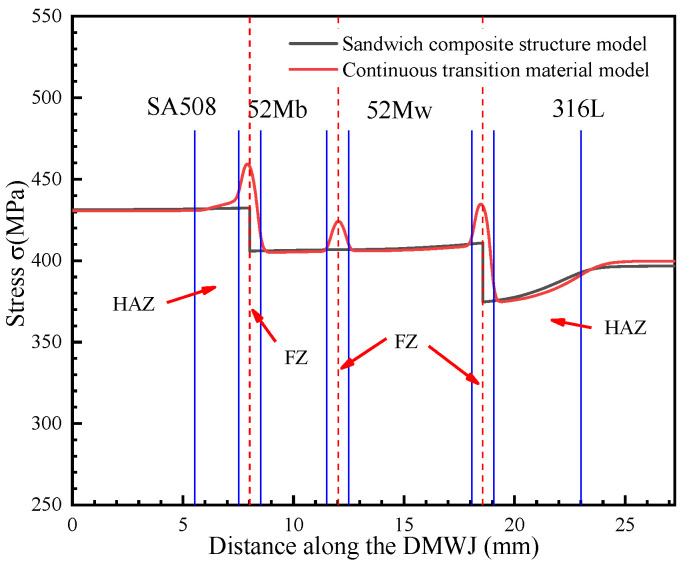
Stress along the DMWJ under an axial load and internal pipe pressure.

**Figure 14 materials-14-04145-f014:**
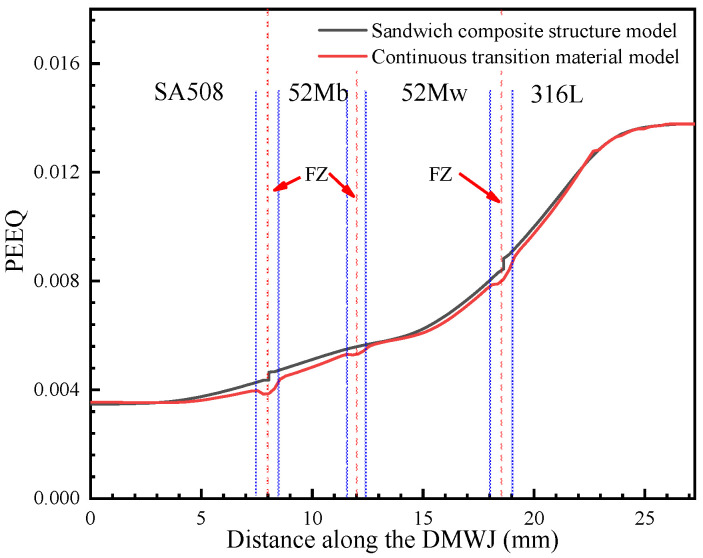
PEEQ along the DMWJ under an axial load.

**Figure 15 materials-14-04145-f015:**
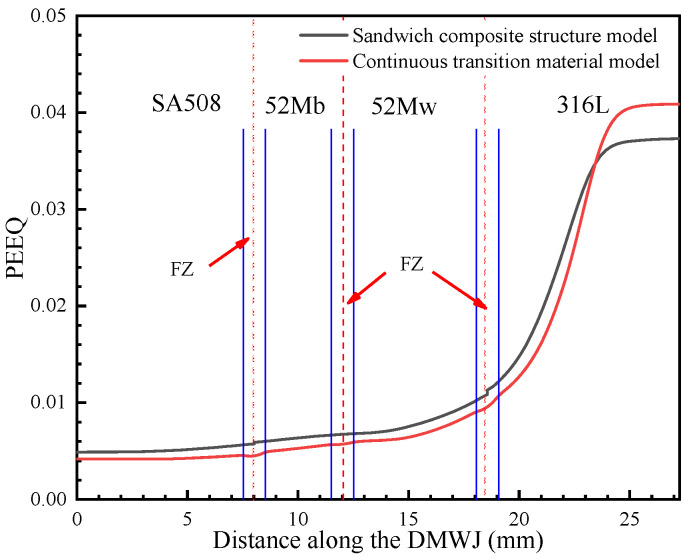
PEEQ along the DMWJ under an axial load and internal pipe pressure.

**Table 1 materials-14-04145-t001:** Chemical composition of materials used in dissimilar metal weld joints (DMWJs) [26].

Material	C	Si	Mn	Cr	S	Ni	Fe	P	Mo	N
SA508	0.170	0.210	1.360	0.16	0.001	0.80	Bal.	0.006	0.490	-
52Mb	0.020	0.110	0.890	29.77	<0.0005	59.20	8.73	0.003	0.008	0.006
52Mw	0.023	0.110	0.900	29.77	<0.0005	59.30	8.74	0.003	0.100	0.006
316L	0.014	0.624	1.576	17.34	<0.001	10.84	Bal.	0.026	2.210	0.116

**Table 2 materials-14-04145-t002:** Mechanical properties of various DMWJ materials at room temperature [20,27].

Materials	Young’s Modulus *E*, MPa	Poisson’s Ratio *v*	Yield Strength $\sigma_{y}$, MPa	Hardening Coefficient $H^{\prime }$
SA508	202,410	0.3	426	1100
Alloy52M	178,130	0.3	400	1000
316L	202,000	0.3	345	850

## Data Availability

The data presented in this study are available on request from the corresponding author.

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
