# Peer review of "Characterization of Mechanical Heterogeneity in Dissimilar Metal Welded Joints"

_materials, 2021, doi:10.3390/ma14154145_

Round 1

Reviewer 1 Report

This paper takes under consideration important topic, namely the modeling of local properties in the welded joints and the influence on overall behaviour of the junction. Proposed solve is interesting, but some remarks must be implemented into the paper:

#44 line - should be "bi-material", not "Bi-material"
#80 line - "to define any mechanical behavior of the material" - Are you sure that "any"? Rather "different"?
#88 line - "calculation of test stress" - How to calculate test stress if it should be assumed or introduced?
#98 and 102 line - there should be shown the matrixes De and Dep
#Figure 1 - what mean the strains ε with upper index "p"? It is not explained. In plastic phase "b" - If the formulas concerning strains were from some paper/research or were developed by author? If were developed - should be explained on which way.
#Figure 2 - The directions "x" and "y" must corresponding to the FE model be placed
#118 line - there are three parts but in text are two
#Table 1 - What is "strengthening factor"? It is "Ultimate strength"? Lack of units of the properties. There should be also the chemical composition of the materials - it will be easier to analyse the results
#138 line - the ">" correct to "≥" (see eq. (1))
#"σy" it is popular mark of "yield strength", not "σo". The marks (indexes) "y" and "0" must be corrected.
#eq. (3) - what is it "σs" and "σs0"?
#Figure 4 - in capition is "hardening coefficient" and at axis is other name and lack of mark H'
#eq. 6 and 7 - there are serious mistakes on boundaries eg. σy=19 = 249 not 365; Hy=23 = 1024 not 850 - these functions are not continuous as is written in text
#186 line - if the model is "axisymmetric" where is axis in the Figure 5? I propose delete this word - it is just simple rectangular two-dimensional model - if I understand, the crossection of the wall
#190 line - there is 4320 units, but model on Fig. 5 has 1080 units (20x54). Where are others? What size has a unit?
#Figure 5 - Lack of dimensions of the model. Where are the zones of weld? This type od constraints cause that this model will be bended (under internal pressure)
#paragraphs 4 and 5 - it is not "veryfication" but rather the comparition of new model with the old. Veryfication should be made on real object by experiment and real measurements of the properties such strain. There are compared the results from old and new model, and the results are better (there are continuous changes) but the values were not veryfied on real object.
#201-202 lines - this sentence suggrests that real curves were analyzed
#205 line - what is "breaking strain"? Is it "strain at failure"?
#Figure 6 and others - If the figures were obtained at load defined by displacemet of 0,5 mm? If yes, thera is any information about "breaking strain". Strain of 3% do not causes the failure in steel.
#Figure 7 - If the figures were obtained at load defined by displacemet of 0,5 mm?
#Figure 8 - pleace to draw the arrows showing the direction of tensile load
#227-228 lines - it is not truth. For both models are present compression and tension stresses.
# What way were the stresses calculated in three directions under two-dimensional model?
#Figure 11- shear stresses are marked by "τ" and the valueas are negligibly small
#which stresses (X, Y and Z) are axial, radial and circumferential - if it is a model based on cylinder
#what is the formula to calculate the equivalent plastic strain PEEQ?
#264-265 lines - "change drasically" or "model fully consideres the difference" - these phreses are very hard - should be more gentle
#271-273 lines - this sentence must be changed (see eg. #Figure 6 and others)
#299 line - "When" - capital letter W

Reviewer 2 Report

The paper is poorly written and many shortcomings have been noticed, as follows:

  1. The title of the paper has to be revised, it contains twice “Mechanical proprieties” and can create confusion.
  2. The abstract has to be re-written because the authors have to briefly describe the knowledge gap, research methodology, the major findings, and several significant conclusions.
  3. The manuscript has to be re-organised following the structure: Abstract, Introduction, Materials and Methods, Results and Discussion, Conclusion.
  4. The originality level of the investigation and the knowledge progress in the dissimilar metal welding field are not demonstrated. There are many articles focused on the mechanical properties of Dissimilar Metal Welded Joint (DMWJ), including of the authors, but the progress beyond the state-of-the-art is not clearly described.
  5. There is no information related to the welding process applied, the main process parameters, etc. The authors study the mechanical properties of dissimilar metal welded joints without taking into consideration the welding process and the thermal effects on the material's behaviour. A coupled thermo-mechanical model has to be developed.
  6. In table 1, “Mechanical properties of various materials at room temperature” are missing the units in the SI system. Moreover, it is strongly recommended to indicate the standards used for describing the mechanical proprieties.
  7. It seems the authors did not consider the influence of temperature on the thermal and mechanical proprieties of materials, which can lead to major errors in the model. Moreover, the authors did not experimentally validate the developed model with their own results. They cite three references from the scientific literature but no discussion is provided.
  8. Figure 5 is only relevant for showing the loadings in the FEM. However, the post-processed result of the FEM analysis should be provided.
  9. Figure 7b x-axis title is True strain, not Stress.
  10. Supplementary discussion on the results, based on scientific arguments, has to be added, in order to explain in detail, the causes that generated the effects. (E.g. Figures 8-15). Moreover, the increase or decrease of the analysed mechanical proprieties have to be compared, and quantitative values should be provided in the discussions.
  11. The discussions on the results have to be moved before in figures/tables/charts (E.g. Figures 8-15).
  12. Because of many errors of grammar and expressions, the translation into the English language must be revised by a native speaker who knows the technical terms, too. Many inadequate and wrong technical terms have been noticed in the manuscript (E.g. Mises stress (von Mises stress), true distance along the DMWJ (distance along DMWJ), etc.)
  13. The conclusions have to highlight the main novel findings and the contribution to the knowledge progress in the approached field.
  14. The references have to be re-ordered and numbered correctly. For example, reference 25 does not exist in the reference list, even if it appears in table 1.

Round 2

Reviewer 1 Report

line #98  - not "nth" but "nth"
Table 1 - wrong value for Ni / 316L
Figure 2 - capital letter in first word
eq. (15) - the equation is the same, but in response is stated some change. In what way was the value change?

Reviewer 2 Report

The manuscript has potential, but many shortcomings have been noticed:

  1. The title of the paper has to be revised. Expressions like “using the user material ….” has to be avoided.
  2. The paper has to be review by a native speaker who knows also the technical terms because it contains many grammatical errors. For example, the first “sentence” of the abstract does not contain any verb.
  3. The authors should use the MDPI Materials Journal MS Word template or LaTeX template to prepare the manuscript and has to be re-organized using the structure: Abstract, Introduction, Materials and Methods, Results, Discussion, Conclusion.
  4. The sentence from lines 51-55 has to be reviewed.
  5. Same comment and suggestion as in 1, to the line 59.
  6. The sentence from lines 60-61 has to be rewritten and proofing tools should be used.
  7. Same comment and suggestion as in 5, to the lines 75-76. “Mechanical behaviour” is repeated.
  8. The authors have to explain the terms from the equations. For example, V from the denominator, in equation 3 is not provided.
  9. The paragraph from lines 121-128 has to be reviewed.  
  10. Standard welding terms and definitions have to be used. Terms like “butting” has to be avoided. In figure 4 the term “welding structure” should be replaced with “welded structure”.  
  11. The phrase from lines 207-210 has to be reviewed.
  12. The phrase from lines 271-277 has to be reviewed: ”... the model developed in this work has a gradual increase…”.  The developed model could not increase, but the results applying the developed model reveal increased stress, etc.
  13. The third-person point of view is generally used in scientific papers. The authors have to rephrase the lines 66, 87, 249, and 322.
  14. The references have to be re-ordered and numbered correctly. For example, reference 31 from line 94 does not exist in the reference list.
